# Integrated Piezoresistive Normal Force Sensors Fabricated Using Transfer Processes with Stiction Effect Temporary Handling

**DOI:** 10.3390/mi13050759

**Published:** 2022-05-11

**Authors:** Ni Liu, Peng Zhong, Chaoyue Zheng, Ke Sun, Yifei Zhong, Heng Yang

**Affiliations:** 1Department of Nephrology, Longhua Hospital, Shanghai University of Traditional Chinese Medicine, Shanghai 200032, China; annie0722@shutcm.edu.cn; 2State Key Laboratory of Transducer Technology, Shanghai Institute of Microsystem and Information Technology, Chinese Academy of Sciences, Shanghai 200050, China; pengzhong@mail.sim.ac.cn (P.Z.); osananajimi@mail.sim.ac.cn (C.Z.); sunke@mail.sim.ac.cn (K.S.); 3University of Chinese Academy of Sciences, Beijing 100049, China

**Keywords:** complementary metal-oxide semiconductor, MEMS, tactile sensor, stiction effect, temporary handling, stiction contact, Au–Si eutectic, flip-chip

## Abstract

Tactile sensation is a highly desired function in robotics. Furthermore, tactile sensor arrays are crucial sensing elements in pulse diagnosis instruments. This paper presents the fabrication of an integrated piezoresistive normal force sensor through surface micromachining. The force sensor is transferred to a readout circuit chip via a temporary stiction effect handling process. The readout circuit chip comprises two complementary metal-oxide semiconductor operational amplifiers, which are redistributed to form an instrumentation amplifier. The sensor is released and temporarily bonded to the substrate before the transfer process due to the stiction effect to avoid the damage and movement of the diaphragm during subsequent flip-chip bonding. The released sensor is pulled off from the substrate and transferred to the readout circuit chip after being bonded to the readout circuit chip. The size of the transferred normal force sensor is 180 μm × 180 μm × 1.2 μm. The maximum misalignment of the flip-chip bonding process is approximately 1.5 μm, and sensitivity is 93.5 μV/μN/V. The routing of the piezoresistive Wheatstone bridge can be modified to develop shear force sensors; consequently, this technique can be used to develop tactile sensors that can sense both normal and shear forces.

## 1. Introduction

Tactile sensation is a highly desired function in the robotics industry [1,2,3,4,5,6,7,8,9]. Makihata [5] recently reported that the requirements for tactile sensors include large-area sensing capability, which requires a large number of sensors, rapid response time, and low cost.

Tactile sensor arrays are also crucial sensing elements in pulse diagnosis instruments [10,11,12,13], which mimic traditional Chinese doctors and determine pulse signals to assess the health of patients. The tactile sensors used for pulse taking must be organized in a large dense array to cover the entire area around the radial arteries, with a sub-millimeter resolution. The integration of sensors with readout circuits is a crucial technology required for large-area, high-resolution sensing.

Extensive research has been conducted to integrate microelectromechanical systems (MEMS) and readout circuits [14,15,16,17,18,19]. Monolithic integration features low-electronic parasitics, reduced chip pinout, and small size. However, the strict thermal budget and process compatibility results in complex processes and performance tradeoff, which present various problems. Hybrid integration [20,21,22], which enables MEMS and complementary metal-oxide semiconductor (CMOS) devices to be optimized independently, is currently the most widely used approach for MEMS and CMOS integration, owing to its short development time, low cost, flexible material selection, and simple fabrication process [23].

Singh et al. [22] proposed a transfer process that can achieve high-density integration by transferring the released MEMS structures onto the readout circuit. Readout circuits can be manufactured using a normal IC foundry and do not undergo etching for release, since the MEMS structures are released before transfer. However, the released MEMS microstructures are movable during the transfer process and can be damaged by shear forces during the bonding and transferring processes. Additionally, the movement of the released MEMS structures decreases the alignment precision of the transferring processes. In a previous study, we proposed a stiction effect temporary handling (SETH) process [24] to temporarily bond the released MEMS structures to the substrates through a stiction effect, which enables temporary handling during the transfer process and reduces alignment errors.

Herein, we present an integrated normal force (force in the z-axis) sensor for pulse diagnosis instruments, wherein the piezoresistive normal force sensor and CMOS readout circuit are integrated using the SETH process. The routing of the piezoresistive Wheatstone bridge can be modified to develop a shear force sensor; consequently, this technique can be used to develop tactile sensors that can sense both normal and shear forces.

## 2. Design and Fabrication

### 2.1. Design of the Integrated Normal Force Sensor

In this study, we primarily focused on normal force sensors, since only the force perpendicular to the sensor surface must be measured for pulse taking. The integrated normal force sensor was fabricated by transferring the released force sensor to a CMOS readout circuit chip, as shown in Figure 1. The normal force sensor, which comprises a diaphragm with piezoresistors installed, was fabricated and released through surface micromachining. The diaphragm was suspended by four beams and temporarily attached to the substrate through the stiction effect [25] of surface micromachining to ensure that the normal force sensor did not move during the transfer process, as shown in Figure 1a. The readout circuit chip comprises two CMOS operational amplifiers, which are redistributed to form an instrumentation amplifier. Pads with amorphous silicon/Ti/Au layers on the surface were fabricated on the chip to serve as anchors for the normal force sensor, as shown in Figure 1b. The normal force sensor was then bonded to the readout circuit chip via Au/Si eutectic bonding, as shown in Figure 1c. The diaphragm was transferred to the readout circuit chip after pulling off from the substrate and breaking the suspension beams, as shown in Figure 1d.

The normal force sensor was designed as the sensing element of the pulse diagnosis instrument, which is used in traditional Chinese medicine. A square, flat diaphragm was employed in the normal force sensor, as shown in Figure 2. The size of the low-stress SiN_x_ diaphragm was 180 μm × 180 μm × 1.2 μm. The polysilicon layer was heavily doped with boron and patterned with piezoresistors and their interconnections. Cr/Pt/Au electrodes were fabricated on top of the polysilicon layer for eutectic bonding and employed as anchors for the diaphragm after transfer. The size of the diaphragm within the electrodes was approximately 120 μm × 120 μm.

Next, the performance of the normal force sensor was simulated. Two piezoresistors were placed perpendicular to the edge of the diaphragm, while two were placed parallel to the edge. Figure 3 depicts the stresses on the piezoresistors, when 500 μN is loaded on the center of the diaphragm. The force 500 μN is equivalent to 260 mmHg pressure on a 120 μm × 120 μm diaphragm, which is slightly higher than the normal blood pressure. The sensitivity of the normal force sensor was calculated to be 34 μV/μN/V, when the longitudinal and transverse gauge factors of boron-doped polysilicon in [26] were used. The normal force sensor is designed to measure pulse signals, whose frequencies are typically lower than 3 Hz. Because the resonant frequency of the sensor is simulated to be as large as 1.11 MHz, as shown in Figure 4, the sensitivities of pulse signals can be considered equal to the DC sensitivity.

The diaphragm was attached to the substrate after release to improve the alignment precision of flip-chip bonding. The temporary bonding strength produced due to stiction must be lower than the bonding strength of the flip-chip to ensure that the normal force sensor can be successfully transferred to the readout circuit chip. Bumps were fabricated under the electrode to decrease the stiction area, as shown in Figure 1a. The total area of the bump surface was designed to be 2869 μm^2^, which was much lower than the area of the electrodes (9792 μm^2^). The normal force sensors can be successfully transferred, even if the temporary bonding strength is equal to the eutectic bonding strength.

To demonstrate integration capability, the normal force sensors were transferred to the CMOS readout circuit chips. The output of the piezoresistive Wheatstone bridge must be amplified using instrumentation amplifiers. Because non-diced wafers of commercial instrumentation amplifiers were unavailable, LMV358 wafers (Yangzhou Genesis Microelectronics Co., Ltd., Yangzhou, China) were employed in our experiments, owing to ease of accessibility. Two LMV358 amplifiers, considered the CMOS version of the LM358 operational amplifier, were redistributed as 2-op amp instrumentation amplifiers [27], as shown in Figure 5. Amplification was determined using external resistors R_1_–R_4_, which presented resistances of 36, 9.1, 9.1, and 36 kΩ, respectively; the amplification was calculated to be 4.96 using the following equation: (1)$$V_{o}=(V_{in2}-V_{in1}(1+\frac{R_{4}}{R_{3}})$$
where *V_in_*_1_ and *V_in_*_2_ are the outputs of the Wheatstone bridge. The 3 dB bandwidth exceeded 100 kHz.

### 2.2. Fabrication

A normal force sensor was fabricated using surface micromachining processes as follows:(a)A 450 nm thick SiO_2_ layer was thermally grown to passivate the substrate. Then, a layer of 800 nm thick low-stress polysilicon was deposited as the sacrificial layer by LPCVD. The polysilicon was subsequently patterned and selectively etched to define the shape of the bumps. Another 200 nm thick layer of low-stress polysilicon was deposited by LPCVD to define the distance between the bumps and substrate. Thereafter, the polysilicon layer was patterned and selectively etched to define the shape of the anchors.(b)A 1 μm thick silicon-rich SiN_x_ layer [28] was deposited by LPCVD to serve as the mechanical layer, tuned to reach a low-residual tensile stress of approximately 50 MPa [29,30]. A 300 nm thick LPCVD polysilicon layer was deposited and heavily doped by boron implantation, followed by patterning and selective etching to form the piezoresisitors. Next, a low-stress SiN_x_ layer of 200 nm thickness was deposited to protect the piezoresisitors.(c)A composite metal layer of Cr/Pt/Au was sputtered and patterned on the piezoresisitors once the contact windows of the piezoresisitors were etched using the RIE technique. The thicknesses of Cr, Pt, and Au were 50, 100, and 300 nm, respectively. The Pt layer of Cr/Pt/Au prevents the Au–Si alloy formed by the subsequent Au–Si eutectic flip-chip bonding process from penetrating the metal pads.(d)The SiN_x_ layer was patterned and selectively etched to form the diaphragm of the tactile sensor and temporarily supported anchors in the silicon nitride diaphragm.(e)The XeF_2_ etching technique was employed to remove the polysilicon sacrificial layer. The released device was subsequently placed in DI water for 24 h and dried at 25 °C for another 24 h to bond the stiction-contact structures temporarily to the substrate using the stiction effect.

Figure 6 illustrates the redistribution flow of the readout circuit chip, as described below:(a)The composite layers of SiO_2_/SiN_x_/SiO_2_ were deposited by PECVD to serve as insulating layers for redistribution. The thickness of each layer was 200 nm. A layer of amorphous silicon of 1 μm thickness was deposited and patterned for subsequent Au–Si eutectic bonding, as shown in Figure 6a.(b)The contact holes were patterned on the insulating layer and the composite metal layers of Ti/Au were sputtered and patterned to redistribute the operational amplifiers to serve as instrumentation amplifiers, as shown in Figure 6b. The Ti layer was used to decompose native oxide on the surface of amorphous silicon during subsequent Au–Si eutectic bonding [31]. The thicknesses of the Ti and Au layers were 50 and 400 nm, respectively.

The released normal force sensor was transferred to the readout circuit chip by flip-chip bonding as follows:(a)The released normal force sensor was bonded to the readout circuit chip using a flip-chip bonder (FinePlacer Lambda, Fintech, Germany), and the temperature, force, and time required for this process were 380 °C, 20 N, and 300 s, respectively.(b)The released normal force sensor was subsequently pulled off from the substrate and broken from the suspension beams by applying a pulling force perpendicular to the bonded device. Because the normal force sensors were released before the transfer process, the readout circuit chips did not undergo release etching. This process demonstrates good CMOS compatibility.

## 3. Results and Discussion

Figure 7a illustrates the released normal force sensor. The interference fingers in the diaphragm and supporting fingers indicated that these structures were temporarily bonded to the substrate due to the stiction effect. The stiction strength during the stiction process was estimated using the longest unattached cantilever to be higher than 7.06 kPa and lower than 22.31 kPa, as shown in Figure 7b.

Figure 8 depicts the integrated normal force sensor, the size of which is approximately equal to those of LMV358, 1070 μm × 640 μm × 525 μm. The maximum alignment error of eutectic bonding was measured to be approximately 1.5 μm, sufficient for the proposed integration process. A Dage Series 4000 bond tester (Nordson DAGE, UK) was used to test the shear strength of Au–Si eutectic bonding. The shear strength of the bonded test structure was approximately 30.74 MPa. The serial resistance of the Au–Si eutectic bonding area is lower than 2 Ω [32], which is much lower than that of polysilicon piezoresistors and can be neglected.

The stress in the transferred normal force sensor caused by the Au–Si eutectic flip-chip bonding process was estimated by comparing the output voltages of the Wheatstone bridge before and after the transfer. The change in the output voltage was in the range of −7.76 to +7.25 mV. Therefore, the stress produced by the Au–Si eutectic flip-chip bonding process was calculated to be in the range of −9.95 MPa to +9.30 MPa, which can be neglected.

The integrated normal force sensor was measured using a set of homemade copper wire weights [24]. The source voltage of the Wheatstone bridge was set to 5 V using Agilent E3631A (Agilent, Santa Clara, CA, USA), and the corresponding output voltage of the instrumentation amplifier was recorded using Agilent 34401A (Agilent, USA), when different masses of the beam-shaped copper wire weights were placed on the diaphragm of the transferred normal force sensor under the microscope; Figure 9 presents the measurement results. The sensitivity was calculated to be 93.5 μV/μN/V. The sensitivity of the piezoresistive Wheatstone bridge was calculated to be 18.8 μV/μN/V at an amplification of 4.96. Nonlinearity was approximately 4%, which was quite large and mainly caused by the uncertainty of the point-of-force application. Five sensors were measured. The deviation of sensitivity was less than 20%. System noise was measured at approximately 200 μV.

Shear force sensors can also be developed with flat diaphragms [33] by modifying piezoresistive Wheatstone bridge routing. When shear force in the x direction is applied on the center of the diaphragm, the left side of the diaphragm moves down while the right side moves up, as shown in Figure 10a. The stress of piezoresistor R_1_ in Figure 10b is tensile, while that of R_3_ is compressive. When the piezoresistors are connected, as in the Wheatstone bridge in Figure 10c, the output is sensitive to shear force and insensitive to normal force. The bumps in the center of the diaphragm can be used as a mesa to improve shear force sensitivity.

## 4. Conclusions

Although the integration of sensors with readout circuits is a crucial technology required for large-area, high-resolution tactile sensing, to date, few integrated tactile sensors have been identified, owing to strict thermal budgets and process compatibility. Tactile sensors are typically integrated with the readout circuit in system levels [1,2,3,4,5,6,7,8,9,10,11,12,13]. In this study, an integrated piezoresistive normal force sensor was presented. The surface micromachined normal force sensor was transferred to the readout circuit chip, with a temporary stiction effect handling process.

The piezoresistive normal force sensor was manufactured using surface micromachining. The readout circuit chip comprised two CMOS operational amplifiers, which were redistributed to form an instrumentation amplifier. The SETH process was used to transfer the released sensor to the readout circuit chip. Because the MEMS structure and readout circuits were manufactured separately, they were optimized independently. Because the MEMS structures were released before transfer, readout circuits did not undergo etching for release and could be manufactured using a normal IC foundry. These processes feature excellent compatibility with IC chips.

The normal force sensor was designed for pulse diagnosis instruments. The size of the transferred normal force sensor was 180 μm × 180 μm × 1.2 μm. The maximum misalignment in the flip-chip bonding process was approximately 1.5 μm. The sensitivity was measured to be 93.5 μV/μN/V. The routing of the piezoresistive Wheatstone bridge can be modified to develop shear force sensors; hence, this technique can be used to develop tactile sensors, capable of sensing both normal and shear forces.

The size of the integrated normal force sensors is approximately equal to those of the readout circuit chips, because the sensors are significantly smaller and sit on top of the readout circuit chips. In our experiments, the wafers of a very-old-version operational amplifier (LMV358) were employed to verify the technology, owing to ease of accessibility. The LMV358 chip size is approximately 1070 μm × 640 μm × 525 μm, and of approximately 0.5 μm minimum line width. Hence, extremely compact sensors can be developed with modern instrumentational amplifiers.

## Figures and Tables

**Figure 1 micromachines-13-00759-f001:**
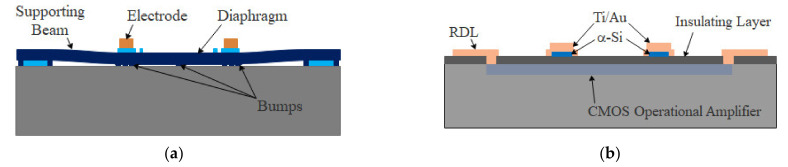
Design process flow of the integrated normal force sensor using stiction effect temporary handling (SETH). (**a**) Normal force sensor is fabricated and temporarily attached to the substrate through the stiction effect; (**b**) readout circuit chip is redistributed, and the pads for eutectic bonding are fabricated; (**c**) normal force sensor is bonded to the readout circuit chip; (**d**) diaphragm is transferred to the readout circuit chip after being pulled off from the substrate and broken from the suspension beams.

**Figure 2 micromachines-13-00759-f002:**
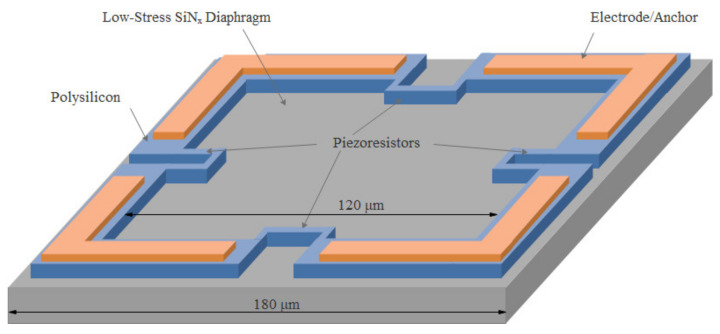
Schematic of the diaphragm of the normal force sensor.

**Figure 3 micromachines-13-00759-f003:**
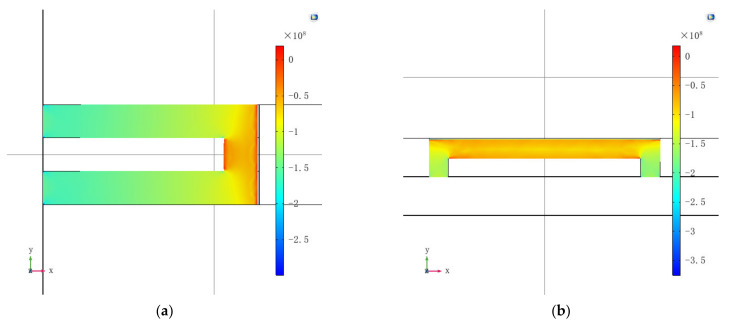
COMSOL simulation results for the normal force sensor. (**a**) Txx along the piezoresistor perpendicular to the edge; (**b**) Tyy along the piezoresistor parallel to the edge.

**Figure 4 micromachines-13-00759-f004:**
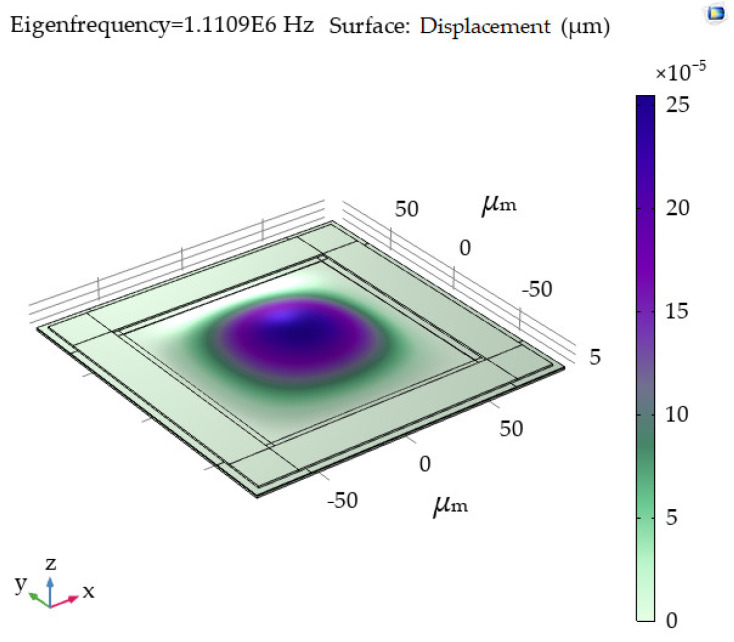
Resonant frequency simulated to be 1.11 MHz by COMSOL.

**Figure 5 micromachines-13-00759-f005:**
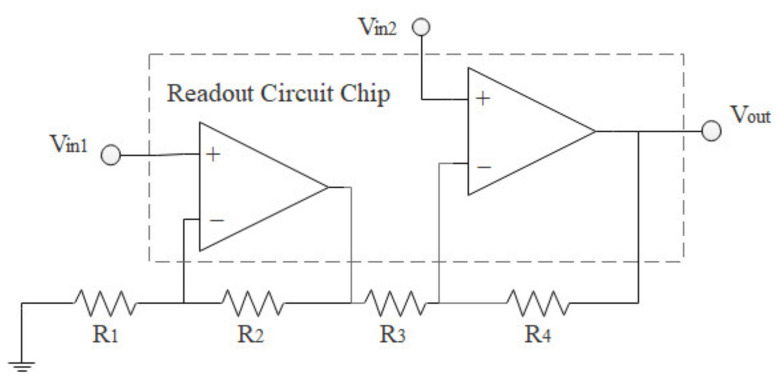
LMV358 is redistributed to serve as a 2-op amp instrumentation amplifier.

**Figure 6 micromachines-13-00759-f006:**
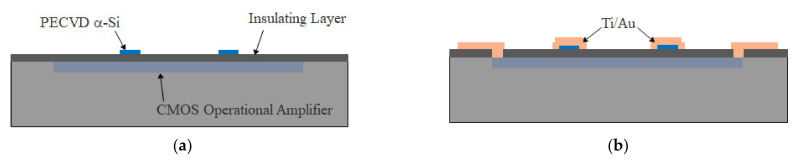
Redistribution flow of the readout circuit chip. (**a**) After composite layers of SiO_2_/SiN_x_/SiO_2_ were deposited by PECVD to serve as insulating layers for redistribution, a layer of amorphous silicon was deposited and patterned for subsequent Au–Si eutectic bonding; (**b**) the contact holes were patterned on the insulating layer, and composite metal layers of Ti/Au were sputtered and patterned for redistributing the operational amplifiers to serve as the instrumentation amplifiers.

**Figure 7 micromachines-13-00759-f007:**
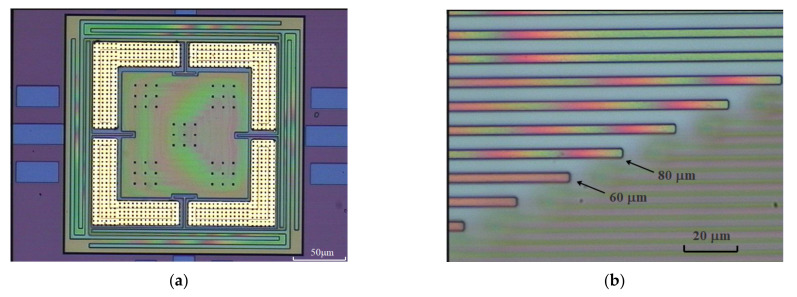
Optical images of released structures; (**a**) diaphragm supported by four beams; (**b**) cantilever array. Cantilevers longer than 80 μm are colorful due to the interference patterns of uneven gaps, while those shorter than 60 μm exhibit uniform color, which indicates that all cantilevers longer than 80 μm adhered to the substrate. Stiction strength is estimated using the longest unstuck cantilever and shortest stuck cantilever.

**Figure 8 micromachines-13-00759-f008:**
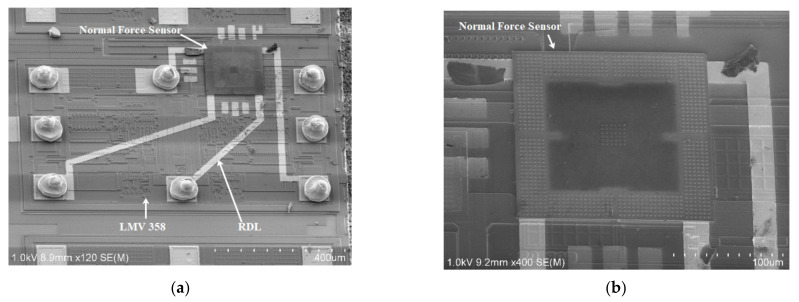
(**a**) Scanning electron micrographs of the integrated normal force sensor; (**b**) close-up view of the transferred diaphragm.

**Figure 9 micromachines-13-00759-f009:**
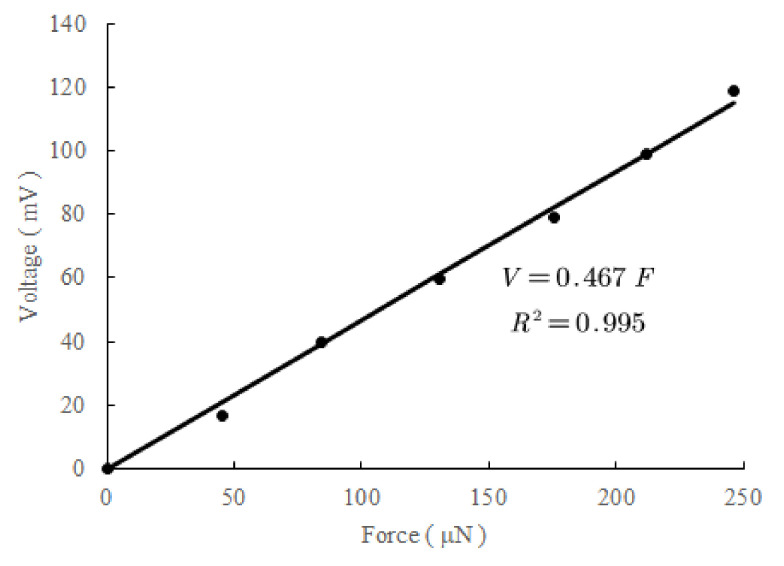
Measurement results of the integrated normal force sensor. Sensitivity was calculated to be 93.5 μV/μN/V.

**Figure 10 micromachines-13-00759-f010:**
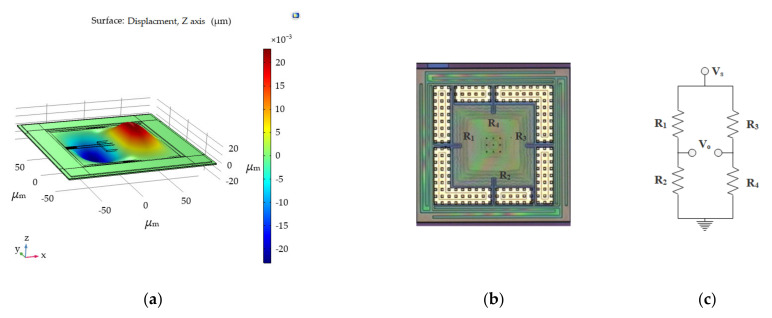
Shear force sensors developed by modifying routing of piezoresistive Wheatstone bridges. (**a**) When shear force in the x direction is applied on the center of the diaphragm, the left side of the diaphragm moves down while the right side moves up. (**b**) Released shear force sensor. (**c**) Wheatstone bridge of the shear force sensor.

## Data Availability

Not applicable.

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
