# Peer review of "Integrated Piezoresistive Normal Force Sensors Fabricated Using Transfer Processes with Stiction Effect Temporary Handling"

_micromachines, 2022, doi:10.3390/mi13050759_

Round 1

Reviewer 1 Report

In this manuscript, the authors demonstrate an integrated piezoresistive normal force sensor manufactured using surface micromachining technology. While the reviewer finds the idea to be interesting, however, the demonstration and level of executing the idea presented in the manuscript does not meet the publishing standard of Micromachines.

Also, the authors may want to address the following comments to improve the manuscript quality:

The title of the second section needs to be corrected. Fig 2(a) mentioned in line 119 is not in the manuscript.

The integration process of integrated normal force sensor is not very clear. The details are suggested.

The detection limit is obtained by simulating the performance of the normal force sensor. However, there is no relevant device for pulse testing. An experiment is suggested.

The manuscript lacks a demonstration of modify the normal force sensor into a shear force sensor by using a Wheatstone bridge.

Reviewer 2 Report

The reported work is of some interest, but further major reversions should be needed before publication.

  1. Please add more descriptions to the picture Caption of Figures, 
    In particular Figure 5, Figure 6 and Figure 7 .
  2. How about the stability and sensitivity of the sensor in Figure 8,is there a standard deviation?
  3. The Discussion Section is oversimplified, should be enhanced, in which practical scenarios, this sensor has potential application possibilities, advantages and so on.. 

Reviewer 3 Report

After reviewing your article, the overall performance is good. More comparison and content in article must be added. The content in this article is too short to expose this issue deeply. The article length must be greater than 12 pages. In Lines 96 and 100, the unit must be confirmed again. “mm” or “um”?

The grammar and content must be revised more. The section 3, results and discussion, can be added more, especially in discussion. The reference number must be added more.

In the technical part, I have some concerns and you must step by step illustrate them.

  1. As you mentioned the CMOS Op. Amp. was applied, please show the amplification gain and the amplification mode (for instance: V/V, I/V, I/I, or V/I). How about the noise figure issue?
  2. In Fig.1, how many times of tactile sensation in this device can be allowed? You need to compare the reliability with the recent commercial products.
  3. In semiconductor manufacturing, SiN layer is usually treated as a high-stress material. However, in Line 138, your mention is low-stress. How do you define this stress in high or low consideration?
  4. In section 3, the results must be compared with the recent published literature or commercial products to expose the high performance of this article. In abstract, the authors mentioned this sensor can sense both normal and shear forces. However, we didn’t see the experimental consequences related to shear force. Please show some shear results and also compare them with the published literature.

5. In Fig.8, the linear regression seems good. Please also show the R2 value in content. If the amplification gain is changed, is the quality of linear regression still great? You should provide the range of amplification gain to fit the request of linear regression, which is strongly related to process deviation.

Round 2

Reviewer 2 Report

The authors have addressed my concerns, the paper could be published.

Reviewer 3 Report

Good!